# INFORMATION-THEORETIC STOCHASTIC CONTRASTIVE CONDITIONAL GAN: INFOSCC-GAN

## ABSTRACT

Conditional generation is a subclass of generative problems when the output of generation is conditioned by a class attributes' information. In this paper, we present a new stochastic contrastive conditional generative adversarial network (InfoSCC-GAN) with explorable latent space. The InfoSCC-GAN architecture is based on an unsupervised contrastive encoder built on the InfoNCE paradigm, attributes' classifier, and stochastic EigenGAN generator. We propose two approaches for selecting the class attributes: external attributes from the dataset annotations and internal attributes from the clustered latent space of the encoder. We propose a novel training method based on a generator regularization using external or internal attributes every $n$-th iteration using the pre-trained contrastive encoder and pre-trained attributes' classifier. The proposed InfoSCC-GAN is derived from an information-theoretic formulation of mutual information maximization between the input data and latent space representation for the encoder and the latent space and generated data for the decoder. Thus, we demonstrate a link between the training objective functions and the above information-theoretic formulation. The experimental results show that InfoSCC-GAN outperforms vanilla EigenGAN in image generation on several popular datasets, yet providing an interpretable latent space. In addition, we investigate the impact of regularization techniques and each part of the system by performing an ablation study. Finally, we demonstrate that thanks to the stochastic EigenGAN generator, the proposed framework enjoys a truly stochastic generation in contrast to vanilla deterministic GANs yet with the independent training of an encoder, a classifier, and a generator. The code, supplementary materials, and demos are available https://anonymous.4open.science/r/InfoSCC-GAN-D113

## 1 INTRODUCTION

Conditional image generation is the task of generating images, based on some attributes. The idea of conditional GAN (cGAN) was proposed by Mirza & Osindero (2014). The authors modified the classic GAN architecture by adding the attribute as a parameter to the input of the generator to generate the corresponding image. They also added attributes to the discriminator input to better distinguish real data. Since then, a lot of other methods have been developed. ACGAN (Odena et al. (2017)) has an auxiliary classifier to guide the generator to synthesize well-classifiable images. ProjGAN (Miyato & Koyama (2018)) improves the approach proposed in ACGAN by utilizing the inner product of an embedded image and the corresponding attribute embeddings. ContraGAN (Kang & Park (2020)) utilizes contrastive $2C$ loss with multiple positive and negative pairs to update the generator.

While these methods have shown impressive results in conditional image generation, they are known to be difficult to train, to lack the image diversity within the same input attribute, and not to always have meaningful data exploration in the latent space.

We propose a new stochastic contrastive conditional generative adversarial network (InfoSCC-GAN) with an explorable latent space. As a baseline generator, we use EigenGAN (He et al. (2021)) generator with interpretable and controllable input dimensions, yet trained in an unsupervised way. EigenGAN ensures that different layers of a generative CNN, controlled by noise vectors, hold different semantics of the synthesized images. EigenGAN is able to mine interpretable and controllable

dimensions in an unsupervised way from different generator layers by embedding one linear subspace with an orthogonal basis into each generator layer. These layer-wise subspaces automatically discover a set of "eigen-dimensions" at each layer corresponding to a set of semantic attributes or interpretable variations, via generative adversarial training to learn a target distribution. By traversing the coefficient of a specific "eigen-dimension", the generator can generate samples with continuous changes corresponding to a specific semantic attribute. The "core" latent space of EigenGAN is generated from the Gaussian distribution. In contrast to that, we use the latent space of the contrastive encoder as the "core" latent space for the generator. By using the contrastive encoder, we have the ability to discover the "inner" attributes from the dataset by clustering the latent space. The "inner" attributes are useful for datasets without external annotations, unbalanced datasets, and datasets with subclasses. By using the encoder, we have an opportunity to compare latent spaces of the real images and the generated images. At the same time, the classifier ensures the correspondence between the attributes of training data and conditionally generated ones. Also, by training the encoder and the classifier independently from the generator, we reduce the general training complexity of the system. It allows avoiding training the encoder and classifier on unrealistic synthetic data, when training it jointly with the generator and discriminator.

The information-theoretical interpretation of the proposed model is provided in Section 2. The experiments and ablation studies are provided in Section 4.

We summarize our contributions as follows:

- We proposed a novel Stochastic Contrastive Conditional Generative Adversarial Network (InfoSCC-GAN) for stochastic conditional image generation with controllable and interpretable latent space. It is based on an EigenGAN, an independent contrastive encoder, and an independent attribute classifier.

- We introduce a novel classification regularization technique, which is based on updating the model with classification loss each $n$-th iteration and updating the generator using the adversarial and classification loss separately.

- We propose a novel method for the attribute selection, based on clustering the embeddings, computed using the pre-trained contrastive encoder.

- We provide an information-theoretic interpretation of the proposed system.

- We perform an ablation study to determine the contribution of each part of the model to overall performance.

## 2 INFORMATION-THEORETICAL FORMULATION

### 2.1 THE TRAINING OF THE ENCODER (STAGE 1)

The encoder training is schematically shown in Figure 1, stage 1. The encoder training is based on the maximization problem:

$$\hat{\phi}_\varepsilon = \underset{\phi_\varepsilon}{\operatorname{argmax}} \, I_{\phi_\varepsilon}(\mathbf{X}; \mathbf{E}), \tag{1}$$

where $I_{\phi_\varepsilon}(\mathbf{X}; \mathbf{E}) = \mathbb{E}_{p(\mathbf{x}, \varepsilon)} \left[ \log \frac{q_{\phi_\varepsilon}(\varepsilon|\mathbf{x})}{q_{\phi_\varepsilon}(\varepsilon)} \right]$, where $q_{\phi_\varepsilon}(\varepsilon|\mathbf{x})$ denotes the encoder and $q_{\phi_\varepsilon}(\varepsilon)$ - the marginal latent space distribution.

In the framework of contrastive learning, (1) is maximized based on the infoNCE framework (van den Oord et al. (2018)). In the practical implementation, one can use approaches similar to SimCLR (Chen et al. (2020)) where the inner product between the positive pairs created from the augmented views originating from the same image is maximized and the inner product between the negative pairs originating from different images is minimized. Alternatively, one can use other approaches to learn the representation $\varepsilon$ such as BYOL (Grill et al. (2020)), Barlow Twins (Zbontar et al. (2021)), etc. without loose of the generality of the proposed approach. It should be pointed out that the encoder is trained independently from the decoder in the scope of the considered setup.

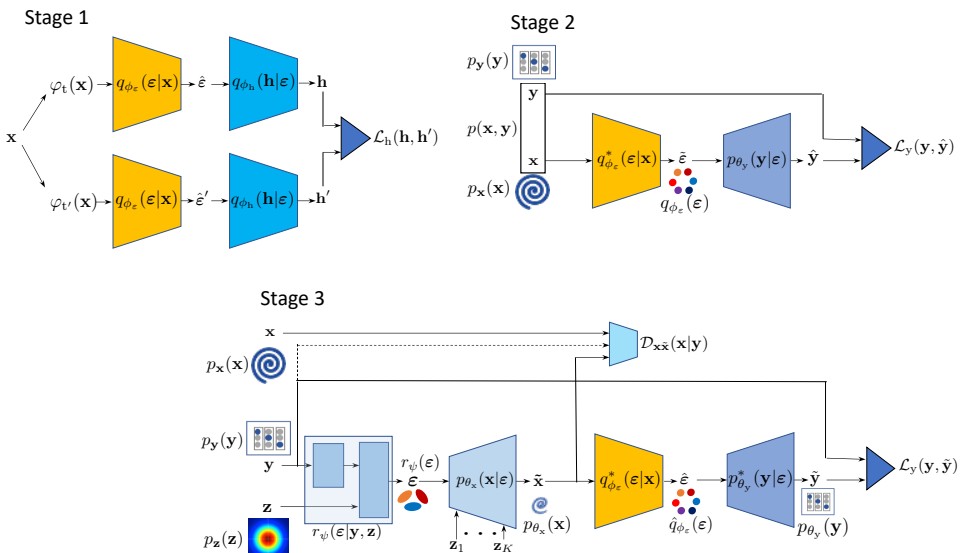

Figure 1: Overview of the proposed InfoSCC-GAN training. Stage 1. Contrastive encoder training. Stage 2. Classifier training. Stage 3. Conditional generator training.

## 2.2 THE TRAINING OF THE CLASS ATTRIBUTE CLASSIFIER (STAGE 2)

The class attribute classifier training is schematically shown in Figure 1, stage 2. The training of the class attribute classifier is based on the maximization problem:

$$\hat{\boldsymbol{\theta}}_{\mathrm{y}} = \operatorname*{argmax}_{\boldsymbol{\theta}_{\mathrm{y}}} I_{\boldsymbol{\phi}_{\varepsilon}^*, \boldsymbol{\theta}_{\mathrm{y}}}(\mathbf{Y}; \mathbf{E}), \tag{2}$$

where $I_{\boldsymbol{\phi}_{\varepsilon}^*, \boldsymbol{\theta}_{\mathrm{y}}}(\mathbf{Y}; \mathbf{E}) = H(\mathbf{Y}) - H_{\boldsymbol{\phi}_{\varepsilon}^*, \boldsymbol{\theta}_{\mathrm{y}}}(\mathbf{Y}|\mathbf{E})$ and $H(\mathbf{Y}) = -\mathbb{E}_{p_{\mathbf{y}}(\mathbf{y})} \log p_{\mathbf{y}}(\mathbf{y})$ and the conditional entropy is defined as $H_{\boldsymbol{\phi}_{\varepsilon}^*, \boldsymbol{\theta}_{\mathrm{y}}}(\mathbf{Y}|\mathbf{E}) = -\mathbb{E}_{p_{\mathbf{x}}(\mathbf{x})} \left[ \mathbb{E}_{q_{\boldsymbol{\phi}_{\varepsilon}^*}(\varepsilon|\mathbf{x})} \left[ \log p_{\boldsymbol{\theta}_{\mathrm{y}}}(\mathbf{y}|\varepsilon) \right] \right]$. Since $H(\mathbf{Y})$ is independent of the parameters of the encoder and classifier, (2) reduces to the lower bound minimization:

$$\hat{\boldsymbol{\theta}}_{\mathrm{y}} = \operatorname*{argmin}_{\boldsymbol{\theta}_{\mathrm{y}}} H_{\boldsymbol{\phi}_{\varepsilon}^*, \boldsymbol{\theta}_{\mathrm{y}}}(\mathbf{Y}|\mathbf{E}), \tag{3}$$

that under the categorical conditional distribution $p_{\boldsymbol{\theta}_{\mathrm{y}}}(\mathbf{y}|\varepsilon)$ can be expressed as the categorical cross-entropy $\mathcal{L}_{\mathrm{y}}(\mathbf{y}, \hat{\mathbf{y}})$.

## 2.3 THE TRAINING OF THE DECODER, I.E., THE MAPPER AND GENERATOR (STAGE 3)

The training of decoder is shown in Figure 1, stage 3. The decoder is trained first to maximize the mutual information between the class attributes $\tilde{\mathbf{y}}$ predicted from the generated images and true class attributes $\mathbf{y}$:

$$(\hat{\boldsymbol{\theta}}_{\mathrm{x}}, \hat{\boldsymbol{\psi}}) = \operatorname*{argmax}_{\boldsymbol{\theta}_{\mathrm{x}}, \boldsymbol{\psi}} I_{\boldsymbol{\psi}, \boldsymbol{\theta}_{\mathrm{x}}, \boldsymbol{\phi}_{\varepsilon}^*, \boldsymbol{\theta}_{\mathrm{y}}^*}(\mathbf{Y}; \mathbf{E}), \tag{4}$$

where $I_{\boldsymbol{\psi}, \boldsymbol{\theta}_{\mathrm{x}}, \boldsymbol{\phi}_{\varepsilon}^*, \boldsymbol{\theta}_{\mathrm{y}}^*}(\mathbf{Y}; \mathbf{E}) = H(\mathbf{Y}) - H_{\boldsymbol{\psi}, \boldsymbol{\theta}_{\mathrm{x}}, \boldsymbol{\phi}_{\varepsilon}^*, \boldsymbol{\theta}_{\mathrm{y}}^*}(\mathbf{Y}|\mathbf{E})$ and $H(\mathbf{Y}) = -\mathbb{E}_{p_{\mathbf{y}}(\mathbf{y})} \log p_{\mathbf{y}}(\mathbf{y})$ and the conditional entropy is defined as $H_{\boldsymbol{\psi}, \boldsymbol{\theta}_{\mathrm{x}}, \boldsymbol{\phi}_{\varepsilon}^*, \boldsymbol{\theta}_{\mathrm{y}}^*}(\mathbf{Y}|\mathbf{E}) = -\mathbb{E}_{p_{\mathbf{y}}(\mathbf{y})} \left[ \mathbb{E}_{p_{\mathbf{z}}(\mathbf{z})} \left[ \mathbb{E}_{r_{\boldsymbol{\psi}}(\varepsilon|\mathbf{y}, \mathbf{z})} \left[ \mathbb{E}_{p_{\boldsymbol{\theta}_{\mathrm{x}}}(\mathbf{x}|\varepsilon)} \left[ \mathbb{E}_{q_{\boldsymbol{\phi}_{\varepsilon}^*}(\varepsilon|\mathbf{x})} \left[ \log p_{\boldsymbol{\theta}_{\mathrm{y}}^*}(\mathbf{y}|\varepsilon) \right] \right] \right] \right] \right]$, where $p_{\boldsymbol{\theta}_{\mathrm{y}}^*}(\mathbf{y} \mid \varepsilon)$ corresponds to the classifier and $q_{\boldsymbol{\phi}_{\varepsilon}^*}(\varepsilon|\mathbf{x})$ denotes the pre-trained encoder. Since $H(\mathbf{Y})$ is independent of the parameters of the encoder and classifier, (4) reduces to the lower bound minimization:

$$(\hat{\boldsymbol{\theta}}_{\mathrm{x}}, \hat{\boldsymbol{\psi}}) = \operatorname*{argmin}_{\boldsymbol{\theta}_{\mathrm{x}}, \boldsymbol{\psi}} H_{\boldsymbol{\psi}, \boldsymbol{\theta}_{\mathrm{x}}, \boldsymbol{\phi}_{\varepsilon}^*, \boldsymbol{\theta}_{\mathrm{y}}^*}(\mathbf{Y}|\mathbf{E}), \tag{5}$$

that under the categorical conditional distribution $p_{\boldsymbol{\theta}_y}(\mathbf{y}|\boldsymbol{\varepsilon})$ can be expressed as the categorical cross-entropy $\mathcal{L}_{\mathrm{y}}(\mathbf{y}, \tilde{\mathbf{y}})$.

Finally, the decoder should produce samples that follow the distribution of training data $p_{\mathbf{x}}(\mathbf{x})$ that corresponds to the maximization of mutual information:

$$(\hat{\boldsymbol{\theta}}_{\mathrm{x}}, \hat{\boldsymbol{\psi}}) = \underset{\boldsymbol{\theta}_{\mathrm{x}}, \boldsymbol{\psi}}{\operatorname{argmax}} \, I_{\boldsymbol{\psi}, \boldsymbol{\theta}_{\mathrm{x}}}(\mathbf{X}; \mathbf{E}), \tag{6}$$

where $I_{\boldsymbol{\psi}, \boldsymbol{\theta}_{\mathrm{x}}}(\mathbf{X}; \mathbf{E}) = \mathbb{E}_{p_{\mathbf{x}}(\mathbf{x})} \left[ \mathbb{E}_{p_{\mathbf{y}}(\mathbf{y})} \left[ \mathbb{E}_{p_{\mathbf{z}}(\mathbf{z})} \left[ \mathbb{E}_{r_{\boldsymbol{\psi}}(\boldsymbol{\varepsilon}|\mathbf{y},\mathbf{z})} \left[ \mathbb{E}_{p_{\boldsymbol{\theta}_{\mathrm{x}}}(\mathbf{x}|\boldsymbol{\varepsilon})} \left[ \log \frac{p_{\boldsymbol{\theta}_{\mathrm{x}}}(\mathbf{x}|\boldsymbol{\varepsilon})}{p_{\mathbf{x}}(\mathbf{x})} \right] \right] \right] \right] \right] = \mathbb{E}_{p_{\mathbf{y}}(\mathbf{y})} \left[ \mathbb{E}_{p_{\mathbf{z}}(\mathbf{z})} \left[ \mathbb{E}_{r_{\boldsymbol{\psi}}(\boldsymbol{\varepsilon}|\mathbf{y},\mathbf{z})} \left[ \mathbb{D}_{\mathrm{KL}}(p_{\boldsymbol{\theta}_{\mathrm{x}}}(\mathbf{x}|\mathbf{E} = \boldsymbol{\varepsilon}) || p_{\boldsymbol{\theta}_{\mathrm{x}}}(\mathbf{x})) \right] \right] \right] - \mathbb{D}_{\mathrm{KL}}(p_{\mathbf{x}}(\mathbf{x}) || p_{\boldsymbol{\theta}_{\mathrm{x}}}(\mathbf{x}))$, where $p_{\boldsymbol{\theta}_x}(\mathbf{x})$ denotes the distribution of generated samples $\tilde{\mathbf{x}}$. Since $\mathbb{D}_{\mathrm{KL}}(p_{\boldsymbol{\theta}_{\mathrm{x}}}(\mathbf{x}|\mathbf{E} = \boldsymbol{\varepsilon}) || p_{\boldsymbol{\theta}_{\mathrm{x}}}(\mathbf{x})) \geq 0$, the maximization of the above mutual information reduces to the minimization problem:

$$(\hat{\boldsymbol{\theta}}_{\mathrm{x}}, \hat{\boldsymbol{\psi}}) = \underset{\boldsymbol{\theta}_{\mathrm{x}}, \boldsymbol{\psi}}{\operatorname{argmin}} \, \mathbb{D}_{\mathrm{KL}}(p_{\mathbf{x}}(\mathbf{x}) || p_{\boldsymbol{\theta}_{\mathrm{x}}}(\mathbf{x})). \tag{7}$$

The above discriminator is denoted as $\mathcal{D}_{\mathbf{x}\tilde{\mathbf{x}}}(\mathbf{x})$. At the same time, one can also envision the discriminator conditioned on the attribute class $\mathbf{y}$ $\mathcal{D}_{\mathbf{x}\tilde{\mathbf{x}}}(\mathbf{x} \mid \mathbf{y})$ that is implemented as a set of discriminators for each subset of generated and original samples defined by $\mathbf{y}$.

## 3 IMPLEMENTATION DETAILS

**Dataset** We test the proposed method on AFHQ (Choi et al. (2020)) and CelebA (Liu et al. (2015)) datasets. AFHQ dataset contains 16130 images belonging to 3 classes: cats, dogs, and wilds animals, CelebA dataset contains 202599 face images with 40 binary attributes. We use AFHQ and CelebA dataset for visual result inspection and AFHQ dataset for the ablations studies.

### 3.1 ENCODER

The proposed encoder is designed to produce the interpretable latent space and it can be used for: (i) internal latent exploration, (ii) feature metric like the VGG-loss (Ledig et al. (2017)), (iii) feature extraction for the classification of the generated samples with "external" and "internal" labels.

We have selected the SimCLR unsupervised encoder since it has shown state-of-the-art performance in unsupervised learning on diverse datasets. By training the encoder in an unsupervised way, it is trained to learn the inner data distribution, which is then used to compare real and generated data. For both AFHQ and CelebA datasets we use Resnet50 (He et al. (2016)) as a base model. We pretrain the SimCLR model for each dataset using contrastive NT-Xent loss (Sohn (2016)). We apply the same augmentations as in the original SimCLR paper. The 2D t-SNE of the extracted features for the AFHQ dataset is shown in Figure 2. The 2D t-SNEs of the extracted features for CelebA dataset for selected attributes are shown in the Appendix A.

### 3.2 CLASSIFIER

The initial idea of using a pre-trained classifier to regularize the generative model is based on the need to generate images that belong to the specific class. Training the classifier jointly with the generator and discriminator requires more time, and is inefficient since in the early iterations the generator network produces poorly generated images that are not similar to the real ones. While it is possible to use L2 or other distance-based metrics to regularize the generator by comparing embeddings between real and generated images computed using the encoder, it requires having predefined pairs, and since our goal is to develop a generative model, we use the pre-trained classifier to regularize the generator. We use a one-layer linear classifier for classification. As an input, we use features extracted using the pre-trained encoder. When training on the AFHQ dataset we use the cross-entropy loss, since each image has one attribute per image, when training on CelebA dataset, we use the binary cross-entropy loss, since each image has multiple attributes per image.

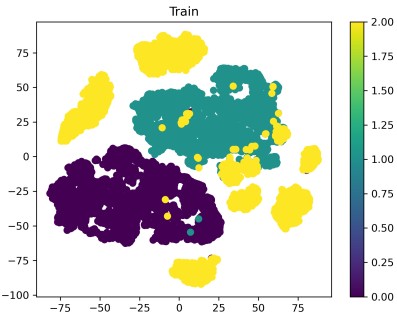

Figure 2: 2D t-SNE of the computed features from the AFHQ dataset. Color represents the class in the dataset: yellow - cat; green - dog; dark violet - wild animal.

### 3.3 Classification regularization

Classification regularization is used to force the generator to generate images conditionally. For every image, that is generated by generator with given input attributes, the embeddings are computed using the pre-trained encoder, and then the embeddings are classified using the pre-trained attributes' classifier. Generator weights are updated so that attributes predicted by the attribute classifier are the same as the input attributes. For the AFHQ dataset with one attribute per image, we use the cross-entropy loss and for the CelebA dataset with multiple attributes per image, we use the binary cross-entropy loss. Unlike other conditional generation methods, which apply classification regularization at each iteration, we apply the classification at each $n$-th iteration, since it allows to balance between the generation of diverse samples and the generation of the samples with specified attributes. The frequency is selected for each experiment with the dataset and attributes. Unlike other methods, where generator weights are updated on all losses at once, we update the generator on each loss separately, so the adversarial loss would not saturate the classification loss.

### 3.4 Clusters

We propose two approaches for selecting the dataset attributes using: (i) "external" attributes either provided with a dataset or annotated manually, (ii) "inner" attributes, assigned by using K-means (Lloyd (1982)) clustering on features extracted using the pre-trained encoder. The 2D t-SNE of the AFHQ dataset features computed using the pre-trained encoder with the "external" attributes from the dataset is shown in Figure 2. There are more than 3 distinct clusters and some of the images from the class of wild animals are semantically closer to the images from other classes. That means that the pre-trained encoder produces semantic similarity. In Figure 3 2D t-SNE features from the AFHQ dataset with the different numbers of the "inner" attributes are shown.

## 4 Experiments

In this section, we describe the experiments and the results obtained for the ablation studies. For the evaluation, we use 3 performance metrics: Fréchet inception distance (FID) (Heusel et al. (2017)), which is used to compare the distribution of generated images with the distribution of real images, inception score (IS) (Salimans et al. (2016)), which is an objective metric for evaluating the quality of generated images and Chamfer distance (Ravi et al. (2020)), which calculates the distance between features of the images. To compute the Chamfer distance, we compute features of the real and generated image by the pre-trained encoder, then compute the 3D t-SNEs of these features, which are used to compute the Chamfer distance. To determine whether the conditional generated images obey the needed attributes, we use attribute control accuracy. The attribute control accuracy is computed as the percentage of the images for which the output of the attribute classifier is the same as an input attribute. The attribute control accuracy measures how good the generator is at conditionally generating samples.

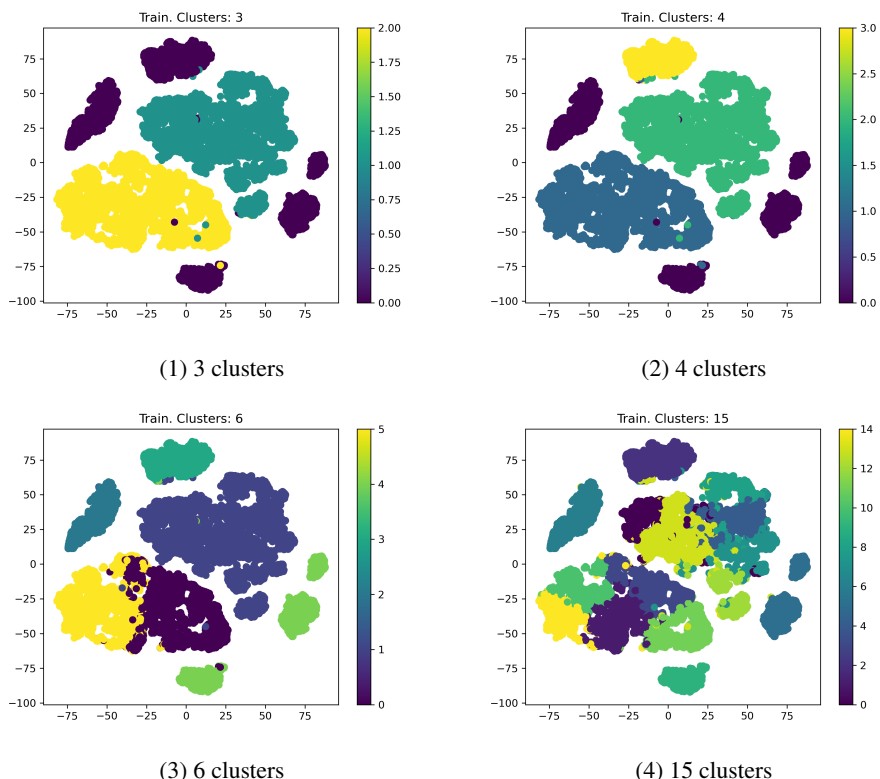

Figure 3: 2D TSNE of AFHQ dataset with different numbers of clusters.

## 4.1 EIGENGAN

We compare the proposed InfoSCC-GAN with the "vanilla" EigenGAN (He et al. (2021)) on the AFHQ dataset. Our model is based on the same generator while using different inputs and conditional regularization. In the current setup, EigenGAN has 6 layers each with 6 dimensions that are used for interpretable and controllable features exploration. The "vanilla" EigenGAN achieves FID score of **29.02** and IS of **8.52** after 200000 training iterations on AFHQ dataset using global discriminator and Hinge loss (Lim & Ye (2017)) (8, 9). The EigenGAN does not allow for interpretable feature exploration for the wild animal images. It can be explained by the imbalance since the "wild" animals class includes multiple distinct subclasses such as tiger, lion, fox, and wolf, which are not semantically close. The interpretable dimensions of the "vanilla" EigenGAN trained on AFHQ dataset are shown in Appendix B. For visualization purposes, we only show the layers and dimensions with visually observable interpretations.

## 4.2 CONDITIONAL GENERATION

We achieve the best FID score of **11.59** and IS of **11.06** using the InfoSCC-GAN approach after 200000 training iterations using Patch discriminator (Isola et al. (2017)) and LSGAN (Mao et al. (2017)) loss (12, 13). In the current setup, we have 6 layers with 6 explorable dimensions. We use 3 cluster classes selected as 3 main clusters using K-means clustering from embeddings computed by the pre-trained encoder. For visualization purposes, the only shown layers are the ones with visually observable interpretations. The interpretable dimensions of the proposed InfoSCC-GAN are shown in Appendix C.

## 4.3 ABLATION STUDIES

In this section, we describe the ablation studies we have performed on the type of discriminator, the discriminator loss, and the number of clusters in the dataset.

Table 1: Discriminator ablation studies

| Discriminator | Loss | FID $\downarrow$ | IS $\uparrow$ | Chamfer distance $\downarrow$ |
|---|---|---|---|---|
| Global | Hinge | 13.08 | 10.71 | 4030 |
| Global | Non saturating | 25.62 | 10.33 | 28595 |
| Global | LSGAN | 29.02 | 9.89 | 45583 |
| Patch | Hinge | 15.95 | 10.51 | 7327 |
| Patch | Non saturating | 14.83 | 10.21 | 5114 |
| Patch | LSGAN | **11.59** | **11.06** | **3645** |

### 4.3.1 DISCRIMINATOR ABLATION STUDIES

In this section, we describe the discriminator and loss ablation studies. For experiments, we use the AFHQ dataset with 3 "inner" clusters. We compare two discriminators: global discriminator and patch discriminator. The global discriminator outputs one value that is the probability of the image being real. The architecture of the global discriminator is inspired by the Eigen-GAN paper. The patch discriminator outputs a tensor of values that represent the probability of the image patch being real, the architecture of the patch discriminator is inspired by the pix2pix GAN (Isola et al. (2017)). We compare these discriminators in combination with discriminator losses: Hinge loss (8,9), non-saturating loss(10, 11) and LSGAN loss(12, 13). The results of the studies are presented in the Table. 1. The visual results are presented in the supplement `https://anonymous.4open.science/r/InfoSCC-GAN-D113/README.md`. The 2D t-SNEs computed from the embeddings are shown in Figure 4. For all of the discriminators and losses, used in the study, the attribute control accuracy is in the range of 99-100%.

In the formulas below $\hat{x} = p_{\theta_\mathbf{x}}(\varepsilon|\boldsymbol{y})$ - is a generated image, $\mathcal{D}_{\mathbf{x}\tilde{\mathbf{x}}}$ - is the discriminator.

The Hinge discriminator loss is defined as following:

$$\mathcal{L}_{\mathcal{D}_{\mathbf{x}\tilde{\mathbf{x}}}} = E_{p_\mathbf{x}(\mathbf{x})}\left[max(0, 1 - \mathcal{D}_{\mathbf{x}\tilde{\mathbf{x}}}(\mathbf{x}))\right] + E_{q_{\phi_\varepsilon}(\varepsilon)}\left[max(0, 1 - \mathcal{D}_{\mathbf{x}\tilde{\mathbf{x}}}(\hat{\boldsymbol{x}}))\right]. \tag{8}$$

The Hinge generator loss is defined as following:

$$\mathcal{L}_{p_{\theta_\mathbf{x}}} = E_{q_{\phi_\varepsilon}(\varepsilon)}\left[max(0, 1 - \mathcal{D}_{\mathbf{x}\tilde{\mathbf{x}}}(\hat{\boldsymbol{x}}))\right]. \tag{9}$$

The non-saturating discriminator loss is defined as following (Lucic et al. (2018)):

$$\mathcal{L}_{\mathcal{D}_{\mathbf{x}\tilde{\mathbf{x}}}} = -E_{p_\mathbf{x}(\mathbf{x})}\left[log(\mathcal{D}_{\mathbf{x}\tilde{\mathbf{x}}}(\mathbf{x}))\right] - E_{q_{\phi_\varepsilon}(\varepsilon)}\left[log(1 - \mathcal{D}_{\mathbf{x}\tilde{\mathbf{x}}}(\hat{\boldsymbol{x}}))\right]. \tag{10}$$

The non-saturating generator loss is defined as following:

$$\mathcal{L}_{p_{\theta_\mathbf{x}}} = -E_{q_{\phi_\varepsilon}(\varepsilon)}\left[\mathcal{D}_{\mathbf{x}\tilde{\mathbf{x}}}(\hat{\boldsymbol{x}})\right]. \tag{11}$$

The LSGAN discriminator loss is defined as following:

$$\mathcal{L}_{\mathcal{D}_{\mathbf{x}\tilde{\mathbf{x}}}} = -E_{p_\mathbf{x}(\mathbf{x})}\left[(\mathcal{D}_{\mathbf{x}\tilde{\mathbf{x}}}(\boldsymbol{x}) - 1)^2\right] + E_{q_{\phi_\varepsilon}(\varepsilon)}\left[\mathcal{D}_{\mathbf{x}\tilde{\mathbf{x}}}(\hat{\boldsymbol{x}})^2\right]. \tag{12}$$

The LSGAN generator loss is defined as following:

$$\mathcal{L}_{p_{\theta_\mathbf{x}}} = -E_{q_{\phi_\varepsilon}(\varepsilon)}\left[(\mathcal{D}_{\mathbf{x}\tilde{\mathbf{x}}}(\hat{\boldsymbol{x}}) - 1)^2\right]. \tag{13}$$

### 4.3.2 NUMBER OF CLUSTERS ABLATION STUDY

First, we compare the performance of the InfoSCC-GAN for the AFHQ dataset with the "external" attributes and with the "inner" attributes of the same number of clusters as classes in "external" attributes. The results of the comparison study are shown in Table 2. The attribute control accuracy metrics are not included in the table, since they are 100% for both types of attributes. As it is shown in Table 2, we achieve better performance with the "inner" attributes for all of the metrics: FID, IS, and Chamfer distance. It can be explained by the fact that when selecting the "inner" attributes the images with the same attributes are semantically closer.

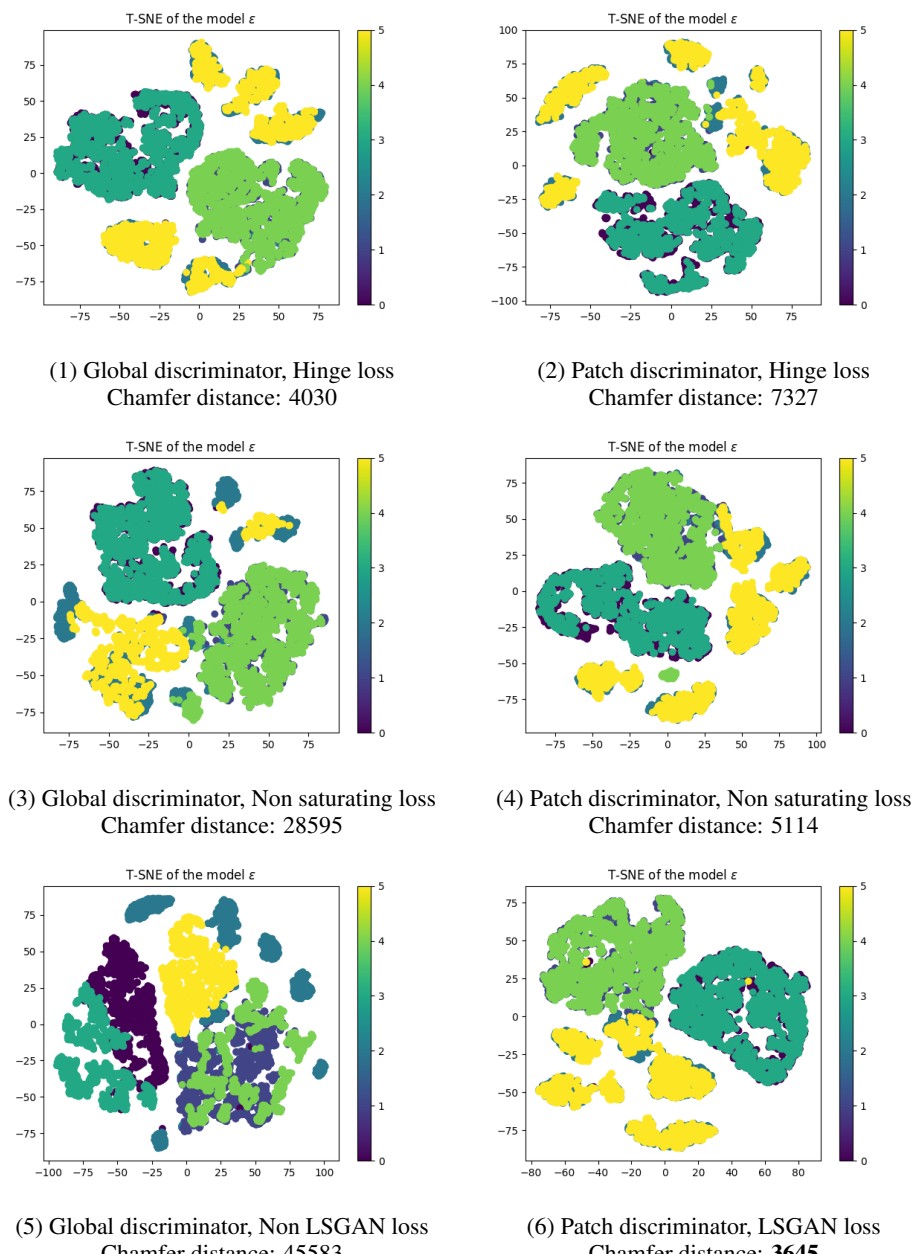

Figure 4: 2D TSNE of AFHQ dataset and conditionally generated samples using different methods. On each plot 0 - cat images from the dataset; 1 - dog images from the dataset; 2 - wild images from the dataset; 3 - generated cat images; 4 - generated dog images; 5 - generated wild images.

Table 2: Comparison of "external" and "inner" attributes

| Attribute type | FID ↓ | IS ↑ | Chamfer distance ↓ |
|---|---|---|---|
| Inner | 13.08 | 10.71 | 4030 |
| External | 14.30 | 10.11 | 4423 |

In this section, we compare generation results on InfoSCC-GAN on the AFHQ dataset, when working with different numbers of clusters. By default, we use 3 clusters. We compare image generation when selecting 4, 6, and 15 clusters. For the experiments, we train the generator with a global dis-

Table 3: Number of clusters ablation study

| Number of clusters | FID ↓ | IS ↑ | Chamfer distance ↓ |
|---|---|---|---|
| 3 | 13.08 | 10.71 | 4030 |
| 4 | 97.37 | 6.77 | 44113 |
| 6 | 148.7 | 5.45 | 94858 |
| 15 | 101.4 | 7.15 | 54016 |

Table 4: Conditional generation results on CelebA dataset with 5 selected attributes

| FID ↓ | IS ↑ | Attribute Control Accuracy ↑ | | | | |
|---|---|---|---|---|---|---|
| | | Bald | Eyeglasses | Mustache | Wearing Hat | Wearing Necktie |
| 27.84 | 9.91 | 93.27% | 99.88% | 95.68% | 94.62% | 98.62% |

criminator, with Hinge loss for 200000 iterations. The comparison results of the numbers of clusters are presented in Table 3, we also compute an attribute control accuracy for each experiment, and for all of them the metrics are 99-100%. The best generation results are achieved when using 3 "inner" classes, which can be explained by the balanced number of samples in each class - approximately 5000 of semantically similar images in each class, since classes were selected by clustering the embeddings.

## 4.4 CELEBA EXPERIMENTS

We run the experiments with CelebA dataset with the different numbers of attributes: 5, 10, and 15 attributes. We have selected the attributes, which can easily and unambiguously be distinguished, so we can perform a visual check of how the proposed conditional generation method performs. For the evaluation, we use the FID score, IS, and attribute control accuracy metrics. We have not used Chamfer distance for the embeddings computed using the pre-trained encoder, since the embeddings are very dense and do not have visible inner cluster structure, but they are still explorable as it is shown in Figure 5. The results on the conditional generation using InfoSCC-GAN with 5 selected attributes are shown in Table 4. When training the model, we have used discrete binary attributes, but we have discovered that the model is able to generate from continuous input attributes from [0, 1] range in an interpretable way, meaning that model learns to apply each attribute to a different degree, depending on the input attribute, if value is closer to 1, the effect of attribute is bigger. This feature is explored in both CelebA demos. We show that proposed InfoSCC-GAN is able to stochastically conditionaly generate samples with multiple attributes. Results with 10 and 15 attributes are shown in the Appendix D.

## 5 CONCLUSIONS

In this paper, we propose a novel stochastic contrastive conditional GAN InfoSCC-GAN, which produces stochastic conditional image generation with an explorable latent space. We provide the information-theoretical formulation of the proposed system. Unlike other contrastive image generation approaches, our method is truly a stochastic generator, that is controlled by the class attributes and by the set of stochastic parameters. We apply a novel training methodology based on using a pre-trained unsupervised contrastive encoder and a pre-trained classifier with every $n$-th iteration using a classification regularization. We propose an information-theoretical interpretation of the proposed system. We propose a novel attribute selection approach based on clustering embeddings computed using an encoder. The proposed model outperforms "vanilla" EigenGAN on AFHQ dataset, while it also provides conditional image generation. We have performed ablations studies to determine the best setup for conditional image generation. Finally, we have performed experiments on AFHQ and CelebA datasets.

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

# A 2D TSNE OF CELEBA FEATURES COMPUTED USING ENCODER

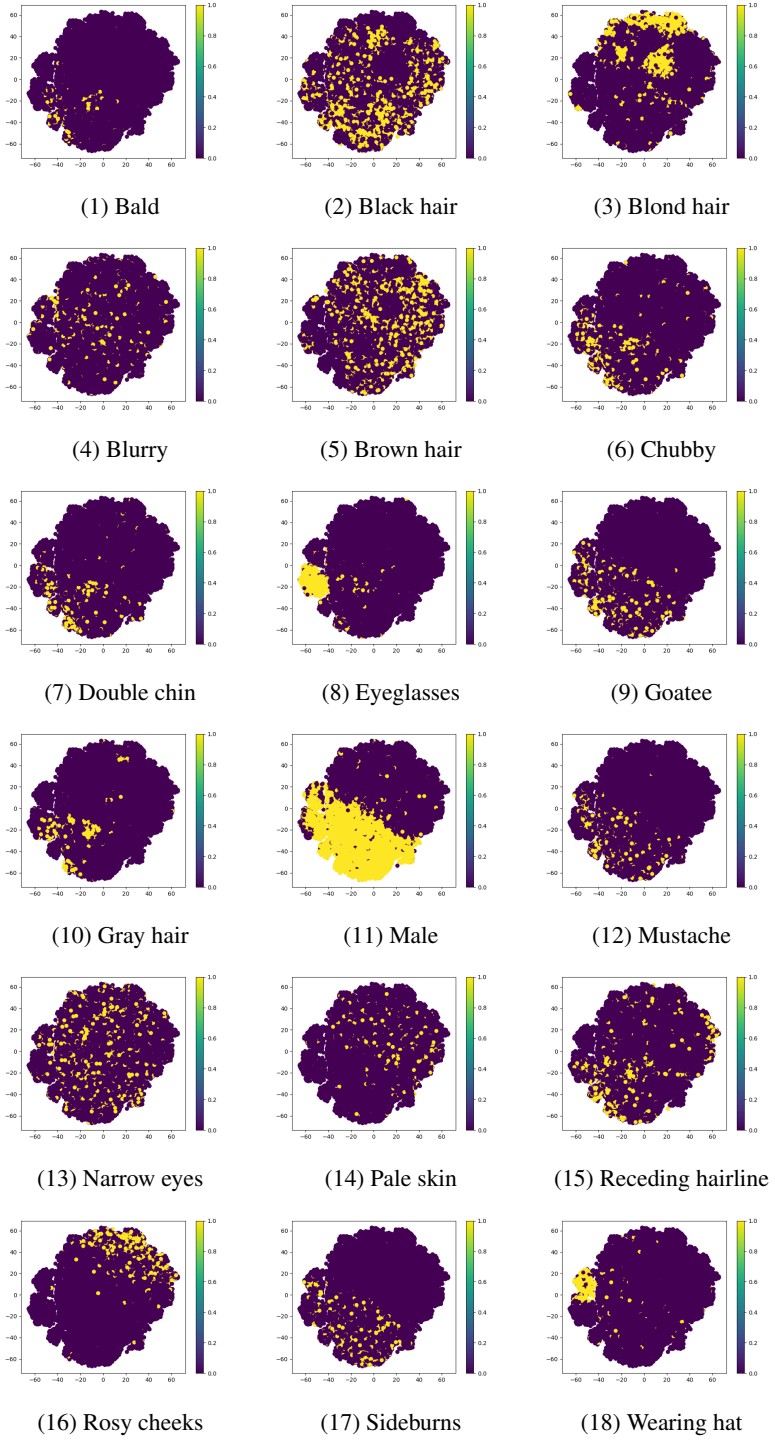

Figure 5: 2D TSNE of CelebA features computed using encoder.

## B "VANILLA" EIGENGAN GENERATION RESULTS ON AFHQ DATASET

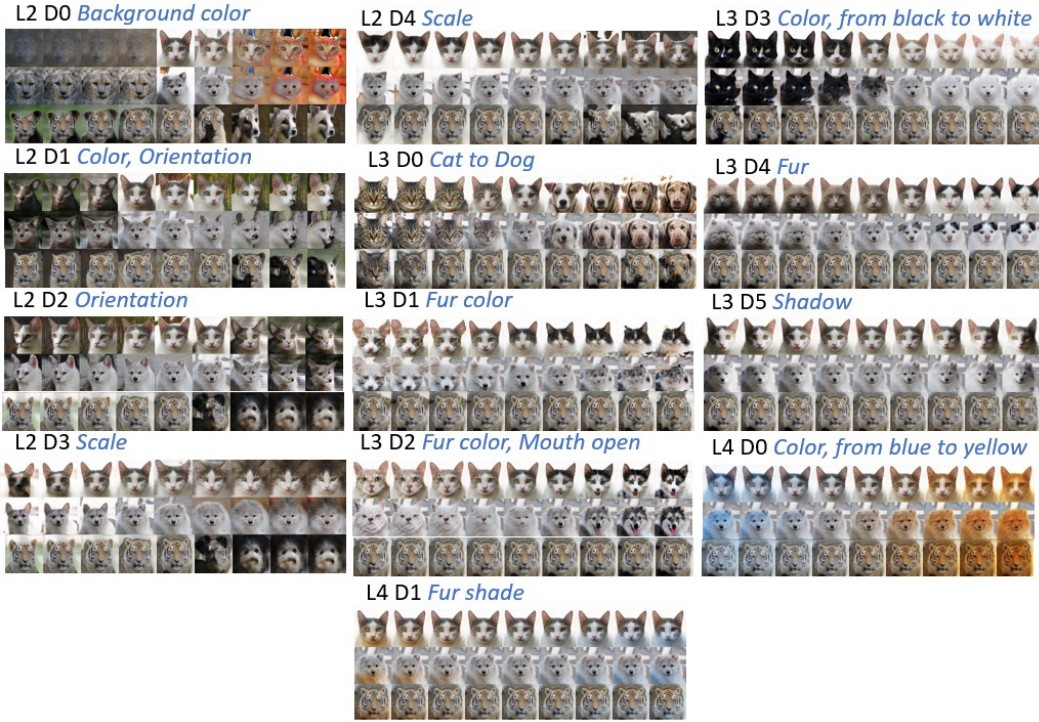

Figure 6: Interpretable dimensions in "vanilla" EigenGAN

## C INFOSCC-GAN CONDITIONAL GENERATION RESULTS ON AFHQ DATASET

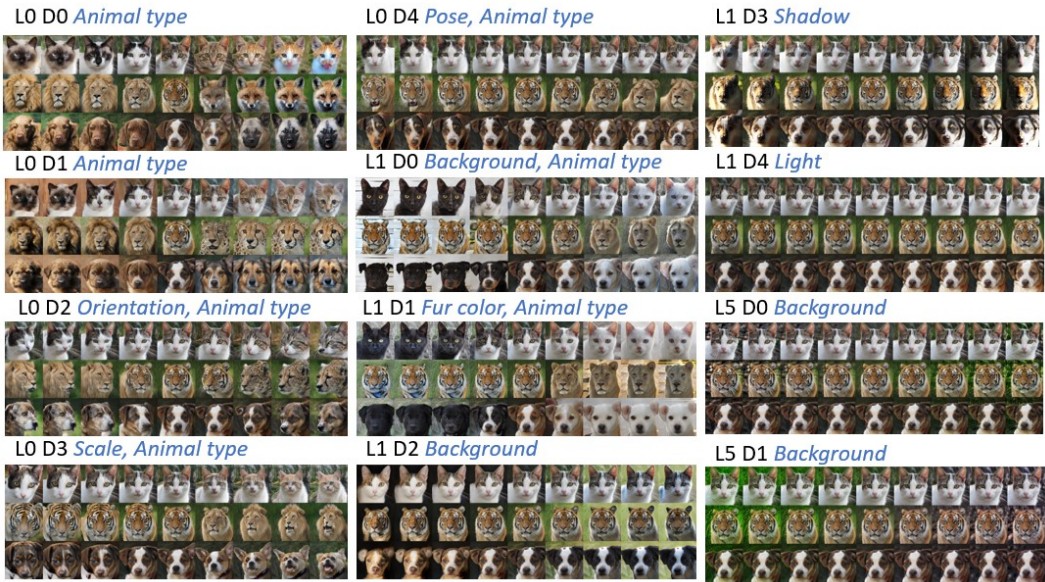

Figure 7: Interpretable dimensions in InfoSCC-GAN

# D  INFOSCC-GAN CONDITIONAL GENERATION RESULTS ON CELEBA DATASET

Table 5: Conditional generation results on CelebA dataset with 10 selected attributes

| FID↓ | IS↑ | Attribute Control Accuracy↑ | | | | |
|---|---|---|---|---|---|---|
| | | Bald | Black Hair | Blond Hair | Brown Hair | Double Chin |
| 32.39 | 9.04 | 89.74% | 89.61% | 86.86% | 85.55% | 84.82% |
| | | Eyeglasses | Gray Hair | Mustache | Wearing Hat | Wearing Necktie |
| | | 99.6% | 81.71% | 92.27% | 92.83% | 89.26% |

Table 6: Conditional generation results on CelebA dataset with 15 selected attributes

| FID↓ | IS↑ | Attribute Control Accuracy↑ | | | | |
|---|---|---|---|---|---|---|
| | | Bald | Blurry | Chubby | Double Chin | Eyeglasses |
| 34.97 | 8.87 | 83.6% | 96.46% | 80.1% | 95.74% | 98.11% |
| | | Goatee | Gray Hair | Mustache | Narrow Eyes | Pale Skin |
| | | 89.09% | 90.78% | 87.64% | 74.22% | 86.91% |
| | | Receding Hairline | Rosy Chicks | Sideburns | Wearing Hat | Wearing Necktie |
| | | 86.46% | 78.88% | 74.9% | 97.64% | 94.87% |

