# OpenReview forum: "Information-theoretic stochastic contrastive conditional GAN: InfoSCC-GAN"
_ICLR.cc/2022/Conference — ICLR 2022 Submitted_

### Official Review · Reviewer_atoU · 2021-11-01

**Correctness:** 2
**Technical Novelty And Significance:** 1
**Empirical Novelty And Significance:** 2
**Recommendation:** 3
**Confidence:** 4

**Main Review:**

Strengths:
1. The proposed method includes a contrastive encoder that can mount existing contrastive losses.
2. For the CelebA dataset, the authors show that InfoSCC-GAN can generate images conditioned on multiple attributes simultaneously.

Weaknesses:
1. The technical novelty of the paper is limited. InfoSCC-GAN introduces a classifier that minimizes cross-entropy, which is the same as AC-GAN [1] or ClusterGAN [2]. I think the difference is the pre-training of the contrastive encoder, which is just an application of self-supervised learning methods.
2. A method for selecting inner attributes using K-means clustering on features already exists [3]. The novelty here is that InfoSCC-GAN pre-trains the encoder via contrastive loss.
3. Major improvements are expected in the experiments section. This work is not the first paper for image generation conditioned on either external and inner attributes. Therefore, more comparative studies are needed to support the advantages of the proposed method. In addition, the paper does not provide generated CelebA images. If space is not enough, t-SNE can be resized or moved to the appendix.
4. The authors mentioned in the introduction that inner attributes are useful for unbalanced datasets. When learning unbalanced attributes on the AFHQ dataset (e.g., the number of clusters = 6), we can expect that the dataset is clustered with six distinct inner attributes (e.g., cat, dog, tiger, lion, fox, wolf). However, FID increases significantly, which means that the image quality is severely compromised. I think there is a contradiction between the authors' argument and the experimental result.

Questions:
1. What is the resolution of the images used in the experiment?
2. In stage 3, the input space of the generator and the output space of the encoder share the same notation ($\epsilon$). Is there any constraint to make the two spaces have the same meaning?
3. When selecting the inner attributes on unbalanced datasets, is $p_y(y)$ determined from a result of clustering?
4. I wonder if the correlation between attributes (e.g., Male & Beard & Bald …) is considered when conditioning multiple attributes in the CelebA dataset. Could you provide any relevant experimental results?

[1] Conditional Image Synthesis with Auxiliary Classifier GANs, ICML 2017.\
[2] ClusterGAN: Latent Space Clustering in Generative Adversarial Networks, AAAI 2019.\
[3] Diverse Image Generation via Self-Conditioned GANs, CVPR 2020.


**Summary Of The Paper:**

This paper proposes a GAN for conditional generation. The authors combine an unsupervised contrastive encoder, stochastic EigenGAN generator, and a classifier. InfoSCC-GAN can perform image generation conditioned on external attributes by maximizing mutual information between input data and class attributes. In addition, clustering on the output space of the pre-trained encoder allows us to generate images conditioned on inner attributes.

**Summary Of The Review:**

In my opinion, this submission is not yet ready for publication. The novelty of the proposed method is limited, and more thorough experiments are needed (Please refer to main review).

------------------------------------------------------------------------------------

Update:

I will keep my score unchanged because the authors have not provided a rebuttal so far.

---

### Official Review · Reviewer_wdWF · 2021-11-01

**Correctness:** 2
**Technical Novelty And Significance:** 1
**Empirical Novelty And Significance:** 1
**Recommendation:** 1
**Confidence:** 4

**Main Review:**

I have multiple major concerns on this paper. First, the idea is not novel, and the mutual information-based objectives do not provide much insights about the idea. Second, this work misses necessary ablations to showcase the advantage of the proposed design. The reported ablation on the discriminator loss and architecture is rather secondary. It is more interesting to see how the classifier design affects the results, e.g., the benefit of having used the SimCLR as the backbone, and only adding the regularization term every $n$ iterations. Third, this work misses comparison with the literature in conditional image synthesis. It is definitely not enough to just compare with EigenGAN. Even the comparison with EigenGAN is very limited. The reported numbers are not strong. For instance, in Tab. 3, the FID drops significantly by changing from 3 to 4 clusters. This indicates that the whole training pipeline is quite fragile.

**Summary Of The Paper:**

The authors aimed to improve EigenGAN for conditional image generation. The conditional information can be obtained via explicit supervision or clustering. The whole idea is not real novel. As already appeared in InfoGAN (2016), a classifier was introduced to facilitate the conditional image generation. Here, the authors suggested to use a pretrained classifier which uses a SimCLR based encoder as the backbone. As the regularization term, the classification loss is active every $n$th iteration. However, in the paper, there are no obvious evidence for the superiority of such design choice.


**Summary Of The Review:**

Based on my main review comments, I suggest "reject".

---

### Official Review · Reviewer_bPdp · 2021-11-03

**Correctness:** 3
**Technical Novelty And Significance:** 3
**Empirical Novelty And Significance:** 3
**Recommendation:** 5
**Confidence:** 4

**Main Review:**

Strengths
1. The authors interpret their proposed method from an information-theoretic perspective, which makes a better theoretical basis for the paper.
2. The authors cluster the latent space embeddings extracted by the encoder as new attribute labels, which is novel.

Weaknesses
1. For classification regularization, the authors didn't indicate the selection method and the precise value of n when applying classification. As you mentioned in section 3.3, “the frequency is selected for each experiment with the dataset and attributes”, does that mean you need to run many times for every dataset and attribute? Since it is one of your contributions, I think more details are needed here.
2. You only compare your method with EigenGAN on the AFHQ dataset, more conditional image generation methods should be compared.
3. I don’t know the purpose of the ablation study on the discriminator. You only compare the existing structure of discriminator and loss functions to choose a better combination, which does not reflect the importance of the components of your framework. I think you should compare the encoder of yours with others to show your superior of the framework.
4. In table 3, the performance is much worse when the cluster is bigger than 3. I think if the encoder produces semantic similarity, then it can cluster better for wild animals when the num is bigger than 3, thus improving GAN training. From Table 3 and Figure 5, I don’t think the encoder produces a good latent space.


**Summary Of The Paper:**

This paper proposes a stochastic contrastive conditional GAN, which consists of an encoder, an attributes’ classifier and a stochastic EigenGAN generator. The encoder learns to extract the features into a latent space based on contrastive learning. It can provide internal attributes of the dataset for training. The classifier is trained to guide the generator to generate images with the corresponding attributes. The EigenGAN generator guarantees to generate stochastic images. Experiments on AFHQ and CelebA datasets show the effectiveness of the proposed method.

**Summary Of The Review:**

Details of the proposed method and experiments are important for readers to follow the main concepts. It would have been better to provide more details in the paper.

---

### Decision · Program_Chairs · 2022-01-20

**Decision:**

Reject

**Comment:**

This paper presents a method for conditional generations for GANs.
The reviewers note the lack of novelty, or the lack of a theoretical or empirical motivation for the novel bits. They point out flaws in the correctness of the paper, and limited experimental evaluation.
The reviewers agree to reject the paper. Unfortunately the authors did not answer the reviewers. I therefore recommend to reject the paper for this conference, and I strongly suggest that the authors address the reviewers concerns if they are to submit this paper again in a future venue.